# Superiority of Mesoporous Silica-Based Amorphous Formulations over Spray-Dried Solid Dispersions

**DOI:** 10.3390/pharmaceutics14020428

**Published:** 2022-02-16

**Authors:** Hongwei Zhang, Minglu Li, Jianmin Li, Anjali Agrawal, Ho-Wah Hui, Demin Liu

**Affiliations:** Bristol Myers Squibb, 86 Morris Ave, Summit, NJ 07901, USA; hz217@scarletmail.rutgers.edu (H.Z.); minglu.li@bms.com (M.L.); jianmin.li@bms.com (J.L.); anjali.agrawal@bms.com (A.A.); ho-wah.hui@bms.com (H.-W.H.)

**Keywords:** mesoporous silica, fenofibrate, spray-dried solid dispersions, stability, manufacturability, in vitro 2-stage dissolution

## Abstract

The aim of this study was to compare the performance of two amorphous formulation strategies: mesoporous silica via solvent impregnation, and solid dispersions by spray drying. Poorly soluble fenofibrate was chosen as the model drug compound. A total of 30% Fenofibrate-loaded mesoporous silica and spray-dried solid dispersions (SDD) were prepared for head-to-head comparisons, including accelerated stability, manufacturability, and in vitro biorelevant dissolution. In the accelerated stability study under 40 °C/75% RH in open dish, mesoporous silica was able to maintain amorphous fenofibrate for up to 3 months based on solid-state characterizations by PXRD and DSC. This result was superior compared to SDD, as recrystallization was observed within 2 weeks. Under the same drug load, fenofibrate-loaded mesoporous silica showed much better flowability than fenofibrate-loaded SDD, which is beneficial for powder handling of the intermediate product during the downstream process. The in vitro 2-stage dissolution results indicated a well-controlled release of fenofibrate from mesoporous silica in the biorelevant media, rather than a burst release followed by fast precipitation due to the recrystallization in the early simulated gastric phase for SDD. The present study demonstrates that mesoporous silica is a promising formulation platform alternative to prevailing spray-dried solid dispersions for oral drug product development.

## 1. Introduction

Poor aqueous solubility is a common challenge for most of the biopharmaceutics classification system’s (BCS) class II or IV drug candidates in the oral route development pipelines. The amorphization of drug molecules is a prevalent strategy to increase solubility and enhance the dissolution rate in the gastrointestinal (GI) tract, subsequently resulting in improved bioavailability [1]. Commonly used techniques for amorphization in the pharmaceutical industry include spray drying [2] and hot melt extrusion [3], however, these conventional techniques may fail in the commercial stage due to multiple limitations, such as economic considerations, being environmentally unfriendly, potential phase separation during long term storage [4], and the difficulty in maintaining amorphous form of drug molecules with low glass transition temperature [5]. 

In the recent decades, loading drug molecules into the mesoporous silica (MS) has been considered as a novel promising alternative to the traditional methods of amorphous formulation development [6]. Credited to their unique characteristics of nanoscale mesopores with high pore volumes and high surface areas, MS materials can effectively entrap drug molecules in the mesopores and suppress the recrystallization by virtue of finite-size effects [7]. As a result, amorphous drugs entrapped in the mesopores can enhance dissolution rate and generate drug supersaturation in aqueous media, leading to a superior bioavailability over the crystalline form [8]. Therefore, MS materials have attracted extensive interest as efficient carriers to deliver poorly aqueous soluble drug molecules. In addition, controlled drug release can be achieved from mesoporous carriers by adjusting key characters of MS materials, such as pore diameter [9], pore volume [10], pore geometry [11], and pore surface functionality [10]. Various feasible loading methods have been explored including solvent impregnation [12], co-spray drying [13], physical mixing [14], microwave irradiation [15], liquid CO_2_ [16], and supercritical CO_2_ [17].

Despite extensive studies in recent years on the structural modification of MS materials, emerging drug-loading methods, and potential applications in drug delivery, there seems to be a lack of a decision as to which processing method can possibly move forward to the commercial stage in comparison with the conventional technology of solid dispersion manufacturing by spray drying or melt extrusion. It has been noted that besides physical mixing, there are many other loading processes that can maintain the homogeneity and amorphous status of drug-loaded MS, among them, solvent impregnation is easier to handle and more economic than the other methods. Additionally, it has also been proved to achieve the greatest increase in drug loading [18]. 

The objective of the current study is to explore the advantages of a mesoporous silica-based amorphous formulation platform over conventional spray-dried amorphous dispersions. In the present study, commercial mesoporous silica material (Parteck^®^ SLC) was used as the carrier, and the Class II drug fenofibrate (abbreviation as Feno, log p = 5.24, aqueous solubility = 0.8 µg/mL) was chosen as the model compound, with poor aqueous solubility and a low melting point [19], to manufacture fenofibrate-loaded mesoporous silica (abbreviation as Feno-MS) via solvent impregnation. Based on a preliminary screening study and past literature [5] the two best-performing polymeric excipients were selected based on stability results (PVP K90 and HPMC E15) to produce two fenofibrate-loaded solid dispersions (abbreviation as Feno-SDD-PVP and Feno-SDD-HPMC), respectively. To our best knowledge, this is the first study where a head-to-head comparison of the performance of an MS-based formulation and the conventional SDD has been performed, including accelerated stability, manufacturability, and in vitro 2-stage biorelevant dissolution.

## 2. Materials and Methods

### 2.1. Materials

Fenofibrate was purchased from Acros Organics (Morris Plains, NJ, USA). The mesoporous silica (Parteck^®^ SLC, disordered pore size at 6 nm) was purchased from Millipore Sigma (St. Louis, MO, USA). The polymers polyvinylpyrrolidone K-90 (PVP K90) and hypromellose E15 (HPMC E15) were purchased from BASF (Florham Park, NJ, USA) and Ashland (Cherry Hill, NJ, USA), respectively. Avicel PH101 (lot# P118832588) was purchased from DuPont (Wilmington, DE, USA). Milli-Q water was obtained from Bristol Myers Squibb (Summit, NJ, USA). All organic solvents (HPLC grade), such as methanol and acetone, were purchased from Sigma-Aldrich (St. Louis, MO, USA). Simulated intestinal fluid (SIF) powder was purchased from Biorelevant.com Ltd. (London, UK).

### 2.2. Methods 

#### 2.2.1. Preparation of Fenofibrate-Loaded Mesoporous Silica

Fenofibrate-loaded mesoporous silica was prepared based on the solvent impregnation method. The initial load of fenofibrate was 10%, 20%, and 30% (abbreviation as 10%-Feno-MS, 20%-Feno-MS and 30%-Feno-MS), respectively. In our preliminary study, the stirring speed of the paddle (IKA NANOSTAR 7.5 digital overhead stirrer, Wilmington, NC, USA) and the loading speed of 50 mg/mL fenofibrate in acetone were optimized at 30% drug load as an upper limit in the study range. Polarized light microscopy (Polarized Light Microscopy (PLM)) was performed after each trial and the preliminary results are concluded in the Appendix A. A stock solution of 50 mg/mL fenofibrate in acetone was added dropwise into the mesoporous silica (Parteck^®^ SLC) at the speed of 2 mL/min using an automatic Masterflex single-syringe infusion pump (Cole-Parmer Inc., Vernon Hills, IL, USA), with paddle stirring at 100 rpm and oil heating at 50 °C. The total solids weight of fenofibrate and mesoporous silica was 50 g. Then, the samples were vacuum-dried at 40 °C for 24 h to remove residual acetone, and all dried samples were stored in 20 mL glass vials with caps sealed at 4 °C.

#### 2.2.2. Preparation of Fenofibrate-Loaded Solid Dispersions by Spray Drying

Fenofibrate-loaded spray-dried dispersions (abbreviation as Feno-SDD) were prepared using a Buchi mini spray dryer (B-290, New Castle, DE, USA). Two polymers, PVP K90 and HPMC E15, were chosen as the polymeric stabilizers, and the initial load of fenofibrate was 30% (abbreviation as 30%-Feno-SDD-PVP and 30%-Feno-SDD-HPMC, respectively). Typically, 3% *w*/*v* of drug–polymeric excipient was dissolved in methanol, which was then spray-dried under nitrogen atmosphere to prevent any potential explosions and drug degradation. Inlet and outlet air temperatures were 100 °C and 50 °C, respectively. The feed pump rate was 10.0 mL/min, and the aspiration rate was 100%. The spray-dried solids were collected and then vacuum-dried at 40 °C for 24 h to remove residual solvent, and the yield of Feno-SDD was 60%. All vacuum-dried SDD samples were stored in 20 mL glass vials with caps sealed at 4 °C.

#### 2.2.3. Accelerated Stability Evaluation

The physical and chemical stability of both Feno-MS and Feno-SDD-PVP were investigated under an accelerated condition of 40 °C/75% RH (relative humidity) in open dish. The samples were pulled out bi-weekly for solid characterizations to check for any physical recrystallization for up to 3 months (3 M). Equivalent to 4 mg of fenofibrate, approximately 40.0 mg of 10%-Feno-MS, 20.0 mg of 20%-Feno-MS, and 13.3 mg of 10%-Feno-MS samples were pulled out from the top, middle, and bottom locations of all Feno-MS samples in the 20 mL glass vials, respectively, for assay and chemical purity tests via a Waters Acquity UPLC system (Waters, Milford, MA, USA). All pulled-out samples were diluted by methanol to 200 μg/mL for UPLC analysis. The system operation, data collection, and processing were performed using the Waters Empower platform. The analytical column was 50 mm × 2.1 mm Acquity UPLC BEH C18, 1.7 µm particle size (Waters, Milford, MA, USA). A gradient method was set up with modification [20]. The two mobile phases were acetonitrile (ACN) and acetate buffer (pH 4.70; 0.01 M), with gradient elution of the ACN phase described as follows: 50% (initial), 70% (1.0 min), 85% (1.4 min), 85% (2.0 min), 50% (2.2 min), and 50% (3.0 min). The injection volume was 1 µL and the flow rate was 0.5 mL/min at ambient temperature with the detection wavelength as 247 nm. The standard stock solution of fenofibrate was accurately prepared at a concentration of 400.0 μg/mL, which was termed as standard 100%. Calibration curves were established over the range of 4.0–400.0 μg/mL, with a regression coefficient greater than 0.999. The nominal retention time of fenofibrate was around 2.0 min.

#### 2.2.4. Solid State Characterizations

##### Powder X-ray Diffraction (PXRD)

A Bruker D8 Advance Diffractometer (Bruker, Billerica, MA, USA) was used to perform PXRD analysis using a Si zero-background holder. A Cu K-α X-ray tube was set at 40 kV and 40 mA, with the wavelength of K_α1_ as 1.5406 Å and the scan speed rate at 0.062° (°2TH)/s from 3° to 40° (°2TH). All samples were tested under ambient conditions.

##### Polarized Light Microscopy (PLM)

An Eclipse LV100N POL polarized light microscope (Nikon Instruments Inc., Melville, NY, USA) was used to detect any birefringent specimens due to fenofibrate crystallization under the polarized light condition. Each PLM image was captured under the auto-exposure background mode at room temperature by 100 times magnification.

##### Thermal Analysis

Differential scanning calorimetry (DSC) thermograms were obtained using a DSC Q2000 mode (TA instruments, New Castle, DE, USA). Calibration was first carried out for checking enthalpy and heating rate using indium and lead. About 3–5 mg of each sample was heated in a crimped aluminum pan at a heating rate of 5 °C/min from 25 °C to 250 °C, with pure nitrogen as a purge gas at 50 mL/min. Thermogravimetric analysis (TGA) thermograms were obtained using a TGA Q5000 mode (TA instruments, New Castle, DE, USA). About 5 mg of each sample in an open platinum plate was heated from 25 °C to 250 °C at 10 °C/min. All DSC and TGA data were analyzed using TA Universal Analysis software.

#### 2.2.5. Scanning Electron Microscopy (SEM)

SEM images of pure crystalline fenofibrate and Feno-MS samples with different drug loads (10–30%) were captured using a Phemon XL Desktop Scanning Electron Microscope (NanoScience Instrument, Phoenix, AZ, USA). Each sample was coated with Au on silicon chips to improve electrical conductivity. The accelerating voltage of the field emission gun was 10 kV and the scale bar was 30 µm.

#### 2.2.6. Particle Size Distribution (PSD)

A laser scattering particle size distribution analyzer (Partica LA-960, Horiba, NJ, USA) was used to measure the particle size distribution of both Feno-MS and Feno-SDD-PVP. About 100 mg of each sample was fed in as dry powder dispersion. Each measurement was conducted in triplicate.

#### 2.2.7. Manufacturability Assessment

The manufacturability of both Feno-MS and Feno-SDD-PVP powders was assessed using the FT4 powder rheometer (Freeman Technology, Tewkesbury, UK) for the following tests: conditioned bulk density (CBD), tapped density (TD), compressibility, and shear cell test. Furthermore, BET surface area of each sample was tested according to USP 43 <846>. The compactibility and compressibility were compared for the binary mixture of microcrystalline cellulose (Avicel PH101) and 30%-Feno-MS, 30%-Feno-SDD-PVP, or pure MS (30:70, *w*/*w*).

##### Conditioned Bulk Density (CBD) and Tapped Density (TD)

Before each test cycle, the sample was conditioned by the standard conditioning process which involves gentle displacement of the powder to loosen and slightly aerate the powder. This process can remove any pre-compaction or excess air. Then, each sample was split to ensure a precise volume of powder by rotating the upper part of the vessel away from the lower part. The powder remaining in the lower vessel was weighed by the FT4 in-built balance to measure the conditioned bulk density. For tapped density, each conditioned sample was consolidated by the tapping mode followed by the splitting process above, and the TD was measured by weighing the remaining powder in the lower vessel.

##### Compressibility Test

In the compressibility test cycle, the change in powder bulk density was measured as a function of applied normal consolidation stress ranging from 0.5 kPa to 15 kPa. The compressibility (CPS%) was calculated as shown by the equation:CPS%=100×(Vc−Vp)/Vp
where *Vc* and *Vp* are the bulk volume before and after the consolidation step, respectively.

##### Shear Test by FT4 Rheometer

The shear test can provide followability information to understand how easily a powder sample previously at rest can flow under normal stress. Four steps, including conditioning, consolidation, pre-shearing, and shearing were included in the test cycle. In the conditioning step, each sample with a uniform and reproducible packing state was established by a helical blade moving downwards and upwards three times at a tip speed of 60 mm/s. During consolidation, a vented piston applied the initial consolidation stress to the sample powder, which was then sheared to achieve a steady state. The shear stress and normal stress were indicated as a pre-shear point. After the pre-shearing procedure, the sample was further sheared to obtain a yield point by lowering the normal stress. In the present study the paired pre-shear/shear procedure was repeated 5 times at normal stress of 6 kPa to obtain a yield locus. A Mohr-circle analysis was plotted to extrapolate the flow function coefficient (ffc).

##### BET Surface Area

The BET surface area of all samples was assessed using a fully automated surface area and porosity analyzer (Micromeritics TriStar II Plus, Norcross, GA, USA). The standard test procedure was followed by USP 43 <846> for each sample. In general, about 0.5–1 g of the sample was weighed in the sample tube. It was then closed with the cap and degassed at 40 °C for 12 h. When degassing was complete, the sample was reweighed and transferred to the analysis port using nitrogen for BET surface area analysis.

##### Compactibility and Compressibility Assessment

Compression profiling was performed on three binary samples by weight ratio as listed in Table 1. To prepare the binary mixture, each component was weighed out and added to a 100 mL amber glass bottle. Mixing was performed using a TURBULA Shaker Mixer T2F (WAB-GROUP, Muttenz, Switzerland) for 10 min at 32 rpm. A binary mixture of 5 g was obtained for each of the three samples following this procedure. True density was measured using a Helium Pycnometer (AcuPyc II, 1340 Pycnometer; Purge Fill Pressure: 19.5 psi; Number of Cycles: 10; Cycle Fill Pressure: 19.5 psi). A 10 mL sample chamber was used, with 2–3 g of sample used for each measurement.

The compact of approximately 200 mg each was obtained using 9 mm diameter round flat tooling by Instron Universal Testing System 5969 (INSTRON, Norwood, MA, USA). The compression force ranged from 1.5 kN to 25 kN. Weight, thickness, and hardness for each compact was measured. The compression profiles, including compactibility (tensile strength vs. porosity) and compressibility (solid fraction vs. compression pressure), of the three binary mixture samples were generated. The compression pressure (MPa), tensile strength (N/mm^2^), solid fraction, and porosity were calculated based on the equations shown below.
$$Compression\;Pressure\;\left(\mathrm{MPa}\right)=\left(1000\times Compression\;Force\;\left(\mathrm{kN}\right)\right)/Diameter^{2}\;\left({\mathrm{mm}}^{2}\right)\;Tensile\;Strength\;\left(\frac{N}{{\mathrm{mm}}^{2}}\right)=\left(2\times Hardness\;\left(\mathrm{kilopond}\right)\times 9.086\right)/(3.14\times Diameter\;\left(\mathrm{mm}\right)\times Thickness\;\left(\mathrm{mm}\right)\;Solid\;Fraction=\left(\left(Weight\;\left(\mathrm{mg}\right)/100\right)/Volume\;\left({\mathrm{cm}}^{3}\right)\right)/\left(True\;Density\;\left(g/\mathrm{mL}\right)\right)\;Porosity=1-Solid\;Fraction$$

#### 2.2.8. Solubility Testing

The solubility of crystalline fenofibrate in simulated gastric fluid (SGF, pH 2.0) and fasted-state simulated intestinal fluid (FaSSIF, pH 6.5) was measured at 37 °C after equilibrium for 24 h. The excessive amount of crystalline fenofibrate was dispersed in relevant media under stirring at 50 rpm for 24 h, and the concentration of crystalline fenofibrate suspension was determined using the Pion MicroFLUX dissolution instrument (Pion, Billerica, MA, USA).

#### 2.2.9. In Vitro 2-Stage Biorelevant Dissolution

The in vitro 2-stage dissolution profiles of the 30%-Feno-MS and 30%-Feno-SDD-PVP were obtained using a Pion MicroFLUX dissolution instrument (Pion, Billerica, MA, USA). Initially, a certain amount of 30%-Feno-MS (or 30%-Feno-SDD-PVP) was added into 7 mL SGF solution under a non-sink condition at 37 °C and stirring speed of 50 rpm. After 20 min, 14 mL FaSSIF solution was added to the vessel, and the theoretical concentration under a sink condition was 20 µg/mL when the total volume in the vessel was 21 mL. The preparation procedures of SGF and FeSSIF were described as the following [21]. The standard calibration curve for SGF and FaSSIF were built with regression coefficients both above 0.99, within the concentration range of 0–25 µg/mL and 0–20 µg/mL, respectively. The concentration of fenofibrate was monitored and recorded at the wavenumber of 294 nm for FaSSIF and 278 nm for SGF per 5 min for up to 4 h. The release data were processed using a second derivative to minimize the interferences of all curves from every measurement time point.

## 3. Results and Discussion

### 3.1. Feno-Loaded Mesoporous Silica at Different Drug Loads

The method of solvent impregnation was chosen to manufacture Feno-loaded MS samples considering its ease of handling and energy-conserving characteristics. Three levels of fenofibrate drug load (10%, 20%, and 30%) were studied, and their physical characteristics and amorphous status were evaluated. As shown Figure 1, all Feno-MS samples with a drug load of 10–30% at time point T0 showed an amorphous halo without any XRD peaks observed in comparison to the XRD diffractogram of crystalline fenofibrate. Moreover, as displayed in Figure 2, neither a significant change in morphology nor obvious fenofibrate on the external surface of MS materials was found when the drug load was increased from 10% to 30% at T0, with SEM images of pure mesoporous silica as the reference. Both results indicated that it was completely possible to encapsulate fenofibrate in an amorphous form in mesoporous silica when drug load was increased up to 30% during the manufacturing process of solvent impregnation. In preliminary experiments, Feno-MS samples at 40% drug load were also prepared, and it was possible to obtain amorphous fenofibrate without any significant diffraction peaks except one small potential peak around 16.7° (2*θ*) through PXRD (Appendix A). However, all attempts failed to obtain amorphous samples of Feno-loaded SDD at 40% drug load by spray drying. Furthermore, 30%-Feno-SDD-HPMC was not amorphous at T0, and 30%-Feno-SDD-PVP failed to maintain the amorphous status of fenofibrate at the time point of 2 weeks (T2W) under 40 °C/75% RH (Figure 1). Thus, only freshly prepared 30%-Feno-SDD-PVP was used for the head-to-head manufacturability and 2-stage dissolution comparison study with 30%-Feno-MS at T0.

As shown in Figure 3, the particle size distribution (PSD) of Feno-MS with 10–30% drug load, 30%-Feno-SDD-PVP, and pure mesoporous silica were also investigated. Though a slight increase was observed in the particle size of 30%-Feno-MS, the PSD profiles of all three Feno-MS samples were comparable with pure mesoporous silica as the reference. In addition, 30%-Feno-SDD-PVP was also within the same particle size range.

### 3.2. Stability Assessment under Accelerated Conditions

The physical and chemical stability profiles were assessed for Feno-MS samples under accelerated conditions of 40 °C/75% RH in open dish for 3 months (T3M). As shown in Figure 1, at 10–30% drug load, no XRD diffraction peaks were detected at the time point of 3M, which indicates that 6-nm-sized mesopores in MS materials successfully restricted the molecular mobility of fenofibrate molecules from recrystallization, and confined fenofibrate molecules in an amorphous form under exposure to high humidity and high temperature. This observation is consistent with SEM images of Feno-MS samples at drug load of 10–30% after 3 months in open dish in accelerated conditions of 40 °C/75% RH as displayed in Figure 2, which showed no significant morphology change in comparison with Feno-MS samples at T0.

The DSC results as shown in Figure 4 agree with the conclusion of PXRD characterization. After 3 months, the accelerated stability study under open dish conditions of 40 °C/75% RH, all the Feno-MS samples at three drug loads (10–30%) showed no endothermic peak as compared with the DSC result of the physical mixture, where an endothermic peak at ~81 °C was found due to the melting of crystalline fenofibrate components. By comparison, 30%-Feno-SDD-PVP failed to maintain an amorphous status of fenofibrate at T2W under the same accelerated conditions, with one endothermic peak, observed at similar position, of fenofibrate melting at ~81 °C due to the recrystallization of fenofibrate molecules.

Moreover, PLM images were captured for Feno-MS samples at a drug load of 10–30% after 3 months’ accelerated conditions of 40 °C/75% RH in open dish, with a physical mixture of 10% *w*/*w* crystalline fenofibrate and 90% *w*/*w* mesoporous silica, pure crystalline fenofibrate, and amorphous mesoporous silica as the control group (Figure 5). PLM, as a polarized microscope method complementary to XRD and DSC, can detect any missed small portion of recrystallized fenofibrate particles with birefringence due to the limit of detection (LOD) of X-ray diffraction and thermal analysis. As displayed in Figure 5, large bright birefringent plates were observed for both pure crystalline fenofibrate and the physical mixture, while no crystalline components were found for all three Feno-MS samples after 3 months of the accelerated stage. The findings of PLM agree with XRD and DSC results that a good control was achieved for Feno-MS samples in preventing recrystallization of fenofibrate against the accelerated scenario, and it also helps rule out the dilemma that recrystallized particles of a small size or at low content might escape detection due to methodological limitations.

So far, multiple research groups have reported the long-term physical stability of MS-based formulations [22]. Superior stability of MS materials over the SDD is credited to the nano size of high volume mesopores, where drug molecules are constrained in a submicrometric space without extra room for the recrystallization [23]. Additionally, potential interaction between drug molecules and silanol groups of MS materials inside the mesopores could contribute to stabilization of the amorphous state as well [24]. Different from the specific structure of MS, SDD is a binary mixture of drug and polymer through spray drying. As evidenced in the Appendix A, weight loss of 30%-Feno-SDD-PVP samples before 100 °C increased from 1.4% to 8.7% after a short-term accelerated open dish stage of 40 °C/75% RH; however, no significant change in weight loss was found for 30%-Feno-MS samples after 3M under the same experimental conditions and drug load by TGA. A lower water uptake of Feno-loaded MS was possibly attributed to the hydrophobic nature of fenofibrate onto the mesopores of silica materials. In comparison, the failure of Feno-loaded SDD in the stabilization of amorphous fenofibrate under the accelerated conditions is possibly due to the moisture absorption of soluble polymer PVPK90 and subsequent phase separation, resulting in drug crystallization being a practical stability challenge for SDD under long-term storage conditions, especially at high temperatures and high humidity [4].

To date, only a few studies have been performed on the accelerated chemical stability of Feno-loaded MS samples, and results were dependent on the type of MS materials [13,25]. To our best knowledge, this is the first report of an accelerated chemical stability study on drug molecule fenofibrate using a novel commercial MS material (Parteck^®^ SLC). As displayed in Table 2, no significant changes in the mean drug assay (*n* = 3) were observed for Feno-MS samples at all three drug loads (10–30%) after 3 months of the accelerated open-dish stage of 40 °C/75% RH. In addition, the three samples for each drug load were pulled out from the top, middle, and bottom locations, respectively, as shown in Figure 6 and the standard deviation of a mean assay was within 2.0 for all Feno-MS samples (Table 2 and Appendix A), which demonstrated good content uniformity within all samples at a drug load of 10–30%. Furthermore, excellent chemical stability was found for all Feno-MS samples without any increase in specific impurities at the retention time (RRT) of 1.9 min or total impurity [20].

### 3.3. Manufacturability Assessment

Besides stability performance, manufacturability is another critical factor to consider whether MS-based oral formulations are amenable to downstream processing steps to manufacture tablets or capsules. The Figure 6 shows the pictures of Feno-loaded powder samples with pure MS as the reference. In the visual inspection, Feno-MS powder looked similar to pure MS, however, Feno-SDD-PVP powder was less dense and very fluffy with a lot of powder sticking onto the wall of vial. Table 3 lists the conditioned bulk density (CBD), tap density (TD), BET specific surface area, compressibility (CPS%), and flow index (ffc) to evaluate the difference in manufacturability between MS-based powder and SDD-based powder. It is apparent that 30%-Feno-SDD-PVP was about 3 times lighter than the MS samples at 20% and 30% drug load, leading to the fluffy issue and major difficulties in handling for downstream steps, including weighing out, screening by sieve, blending, lubrication, etc. Even though the BET specific surface area of Feno-MS material decreased by about 3–4 times after the mesoporous pores were loaded with fenofibrate, the BET value was still 2 orders of magnitude higher than that of Feno-SDD-PVP samples under the same drug load. The BET results indicate that the drug release mechanism between MS and SDD might be different. MS formulation provides a porous structure with a high surface area for confined drug molecules to release [22], but SDD formulation is a miscible or partially miscible powder with polymers dissolved or swollen to facilitate drug release [26]. The CPS (%) of 30%-Feno-SDD-PVP was much higher than MS-based materials, which indicated that the proportional reduction in the thickness of SDD-based material under the prescribed pressure from 0.5 kPa to 15 kPa was significantly large. A large CPS can potentially result in a very dense compact during dry granulation or the tableting process. Finally, the very low ffc values seen in the SDD sample indicate the cohesive nature of the powders after spray drying; however, both 20%-Feno-MS and 30%-Feno-MS showed free-flowing characteristics after the loading of fenofibrate as compared to pure MS. Overall, Feno-MS samples at 20% and 30% drug load both showed much better powder handling and flowability than 30%-Feno-SDD-PVP powder obtained through spray drying, and the latter requires the addition of excessive flow-enhancing excipients in the downstream process.

To mitigate the interference of the material’s characteristics, such as cohesiveness and static charge, 30%-Feno-MS, 30%-Feno-SDD-PVP, and pure MS were blended with filler MCC PH101 at the same weight ratio of 30:70. The three binary mixture samples were evaluated by compression profiling. As shown in Figure 7, similar acceptable compactibility was obtained for both binary mixture samples, indicating that both formulations can be compressed into a slug or a tablet under target tensile strength or hardness. The compressibility results were consistent with previous studies of CPS (%) by FT4, in which an SDD-based binary mixture was easier to be compressed with a large decrease in volume under specific pressure as compared to an MS-based sample. The results agree with other research that elastic fillers such as microcrystalline cellulose and disintegrants should be appropriately considered for the downstream process development of MS-based formulations [28].

### 3.4. In Vitro 2-Stage Biorelevant Dissolution Study

In vitro 2-stage dissolution profiles of Feno-MS and Feno-SDD-PVP samples are compared in Figure 8. The Feno-SDD-PVP sample showed a burst release of fenofibrate at the first 5 min time point in the SGF stage, followed by a sharp drop in drug concentration to the equilibrium solubility level of 9.3 µg/mL at the 20 min time point when gastric stage 1 was transferred into stage 2 with FaSSIF media. The concentration of fenofibrate from SDD-based samples further dropped down to about 4.2 µg/mL at the first 5 min of the FaSSIF stage, then the concentration slowly increased again to 12.1 µg/mL at 3 h. The Feno-SDD-PVP sample dissolution profile deviated from the “spring-and-parachute” effect of amorphous solid dispersions, with a rapidly dissolving and supersaturating “spring” observed without a precipitation-retarding “parachute” [29]. It is indicated that polymer PVPK90 failed to retard the recrystallization of supersaturated amorphous fenofibrate, resulting in crystallized fenofibrate with low soluble exposure when the buffer stage was transferred from SGF to FaSSIF. In comparison, Feno-MS has achieved a well-controlled release of fenofibrate in the 2-stage dissolution test, with the drug concentration gradually increasing when the buffer changed from the gastric stage to the fasted intestinal stage. Credited to the nanoscale mesopores that can restrain the amorphous fenofibrate molecules from premature release in the gastric stage, the Feno-MS sample successfully escaped from the fate of a rapidly dissolving and supersaturating “spring” without a safe “parachute”, as seen in Feno-SDD sample.

To date, the drug release mechanism of mesoporous silica material is still unclear with contradicting opinions. Early suggestions indicate that the release behavior of a drug-loaded mesoporous system can follow a classic Higuchi equation to consider the effect of porosity, drug load, drug solubility, and the diffusion coefficient in the silica matrix [11,30,31]. However, some other studies have reported that the Higuchi equation is not applicable to drug-loaded MS due to the deviation from the expected drug release profile, and instead, two-step drug release has been proposed consisting of an initial burst release from the physically entrapped drug, and a slow release of the drug chemically bonded to the silica surface in the second step [22,32]. Interestingly, the present 2-stage in vitro dissolution study showed no burst release in first 20 min of SGF stage, but did show gradually increasing release behavior for the 30%-Feno-MS sample in the second stage after FaSSIF was added. It was possibly the presence of hydrophobic fenofibrate molecules that prevented the initial burst release, but continuous hydroxylation on the silica surface when in contact with SGF aqueous medium could result in a more hydrophilic environment in the mesopores that facilitated the drug release later in the second stage [33]. In addition, more physiological surfactants, including bile salts and lecithin, were introduced in second dissolution stage after FaSSIF was added, which could effectively increase the solubility and lead to a persistent, increasing supersaturation. To better understand the drug release mechanism, a study on the potential interactions between specific functional groups of fenofibrate and the silanol groups of silica surfaces is required in the future, in order to investigate drug release behaviors caused by chemical interactions such as hydrogen bonding.

## 4. Conclusions

In the present study, a head-to-head comparison of stability, manufacturability, and release in 2-stage biorelevant media was performed between the two-formulation platform. Interestingly, the accelerated stability of mesoporous silica performed excellently under 3 months of the accelerated stage of 40 °C and 75% RH, while no spray-dried solid dispersion samples survived for more than 2 weeks without recrystallization. In addition, mesoporous silica can maintain good flowability after the loading of fenofibrate. A good flowability mitigates the challenges associated with handling fluffy and sticky powder as seen in the solid dispersions. More importantly, the in vitro 2-stage biorelevant dissolution study showed no burst release of fenofibrate in SGF, and a slow release in FaSSIF for Feno-loaded mesoporous silica as compared to spray-dried solid dispersions, which indicated a well-controlled release capability based on the platform of mesoporous silica. In conclusion, this study supports the use of mesoporous silica as an alternative approach for oral amorphous formulation development, with its superior stability, manufacturability, and in vitro release performance compared to conventional spray-dried solid dispersion.

## Figures and Tables

**Figure 1 pharmaceutics-14-00428-f001:**
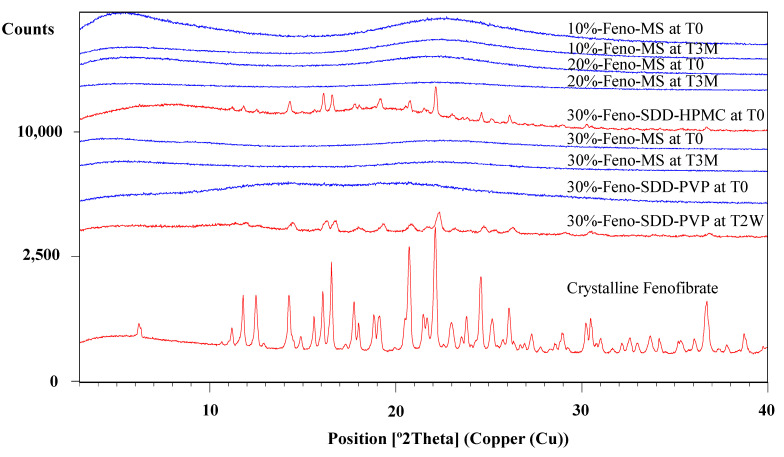
The PXRD overlay of Feno-MS samples (10%, 20%, and 30% drug load at T0 and T3M under accelerated conditions of 40 °C/75% RH in open dish), Feno-loaded SDD samples using polymers PVP and HPMC (30% drug load at T0 and T2W under the same accelerated conditions), and crystalline fenofibrate.

**Figure 2 pharmaceutics-14-00428-f002:**
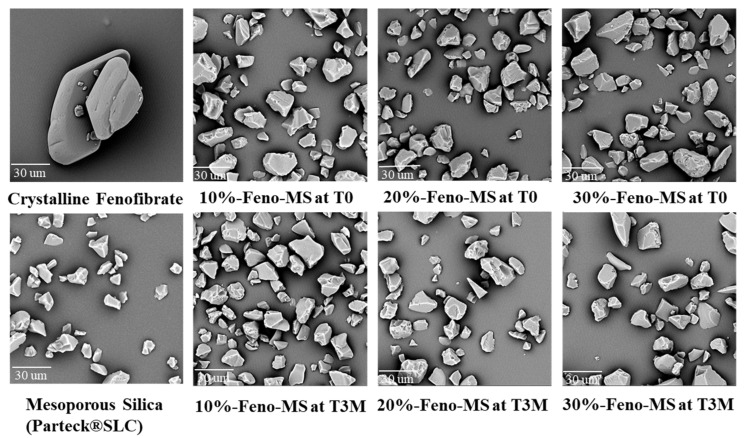
SEM images of Feno-MS samples (10%, 20%, and 30% drug load at T0 and T3M under accelerated conditions of 40 °C/75% RH in open dish), crystalline fenofibrate, and mesoporous silica (Parteck^®^ SLC).

**Figure 3 pharmaceutics-14-00428-f003:**
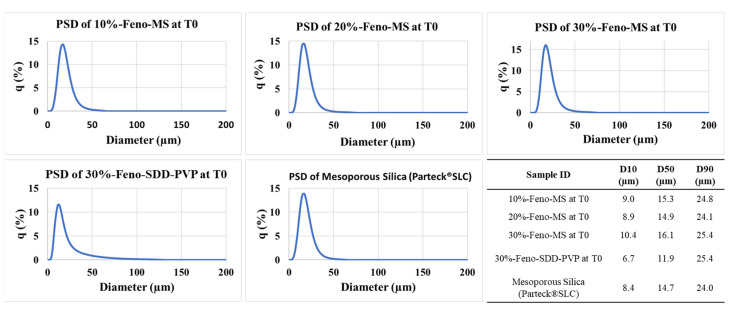
PSD profiles of Feno-MS samples at drug load of 10%, 20%, and 30% at T0, 30%-Feno-SDD-PVP at T0, and mesoporous silica (Parteck^®^ SLC).

**Figure 4 pharmaceutics-14-00428-f004:**
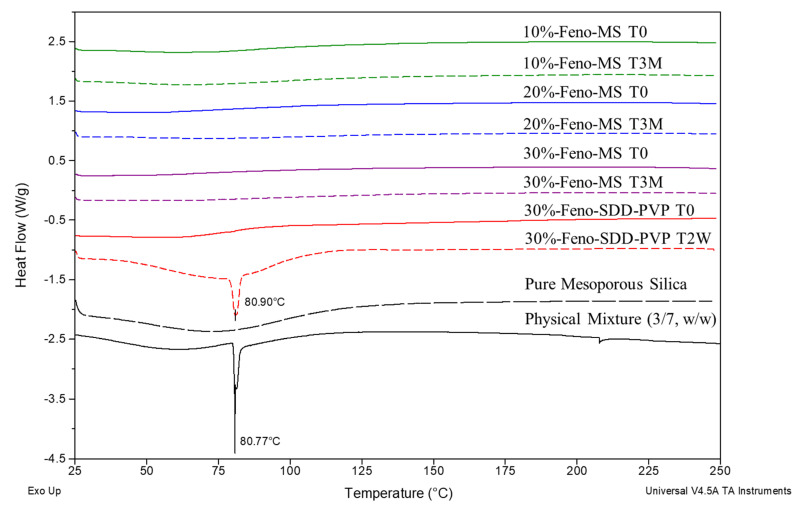
The DSC overlay of Feno-MS samples (10–30% drug load at T0 and T3M under accelerated conditions of 40 °C/75% RH in open dish), Feno-SDD-PVP samples (30% drug load at T0 and T2W under the same accelerated conditions), pure mesoporous silica (Parteck^®^ SLC), and the physical mixture of crystalline fenofibrate and mesoporous silica at a weight ratio of 3:7 (*w*/*w*).

**Figure 5 pharmaceutics-14-00428-f005:**
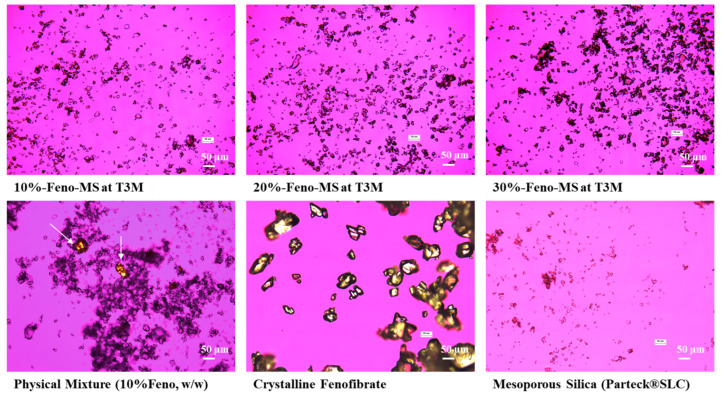
PLM images of Feno-MS samples (10%, 20%, and 30% drug load at T3M under accelerated conditions of 40 °C/75% RH in open dish), the physical mixture of crystalline fenofibrate and mesoporous silica at a weight ratio of 10:90 (*w*/*w*), pure crystalline fenofibrate, and mesoporous silica (Parteck^®^ SLC). Magnification × 100.

**Figure 6 pharmaceutics-14-00428-f006:**
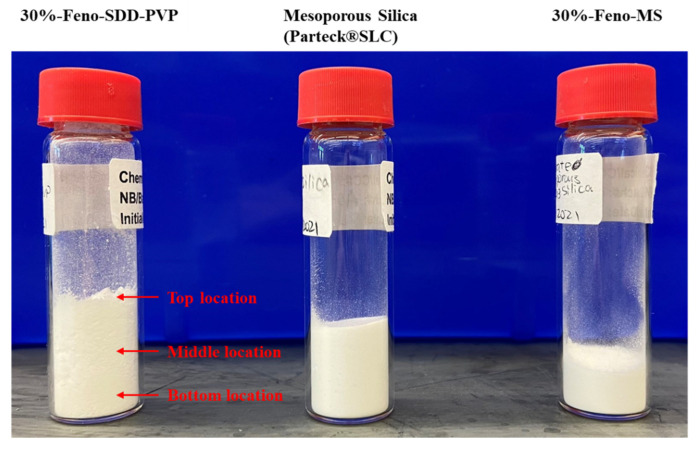
Pictures of 30%-Feno-SDD-PVP (left), mesoporous silica (Parteck^®^ SLC, middle), and 30%-Feno-MS (right). Three samples were pulled out from top, middle, and bottom locations for assay and impurity tests of Feno-SDD and Feno-MS samples.

**Figure 7 pharmaceutics-14-00428-f007:**
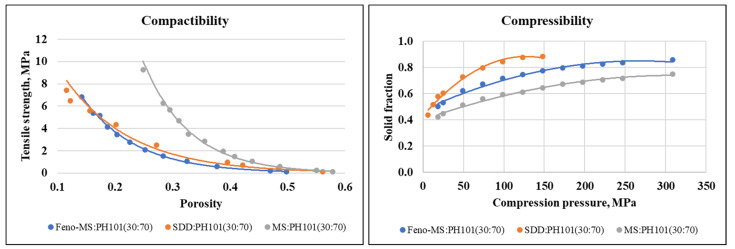
The compactibility and compressibility profiles of blending a mixture of MCC PH101 and three different powder samples at a weight ratio of 30:70 (*w*/*w*): 30%-Feno-MS (blue), 30%-Feno-SDD-PVP (orange), and pure MS (grey).

**Figure 8 pharmaceutics-14-00428-f008:**
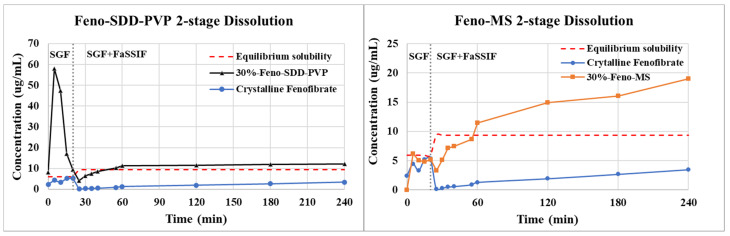
The 2-stage dissolution profiles of 30%-Feno-SDD-PVP and 30%-Feno-MS with crystalline fenofibrate as the reference. SGF stage: 0–20 min; SGF + FaSSIF stage: 20–240 min.

**Table 1 pharmaceutics-14-00428-t001:** Materials used in compression profiling.

Material	Lot of Three Binary Samples	Amount (g)
**Feno-SDD/Avicel PH101 Binary Mixture**		
30%-Feno-SDD-PVP	N/A	1.5
Avicel PH101	P118832588	3.5
**Feno-MS/Avicel PH101 Binary Mixture**		
30%-Feno-MS	N/A	1.5
Avicel PH101	P118832588	3.5
**MS/Avicel PH101 Binary Mixture**		
Mesoporous Silica	N/A	1.5
Avicel PH101	P118832588	3.5

**Table 2 pharmaceutics-14-00428-t002:** Assay and chemical impurity of the Feno-loaded MS or SDD samples at the initial and 3M accelerated open dish stage of 40 °C/75% RH.

Sample ID	Fenofibrate Drug Load (%)				Specific Impurity RRT 1.9		Total Impurity	
	Initial		3 Months at 40 °C/75% RH, Open Dish		Initial	3 Months at 40 °C/75% RH, Open Dish	Initial	3 Months at 40 °C/75% RH, Open Dish
	Mean (*n* = 3)	SD	Mean (*n* = 3)	SD				
10%-Feno-MS	9.9	0.8	9.8	1.2	ND	ND	ND	ND
20%-Feno-MS	20.9	2.0	21.1	0.4	ND	ND	ND	ND
30%-Feno-MS	32.6	1.2	33.1	0.8	ND	ND	ND	ND
30%-Feno-SDD-PVP	29.4	3.9	N/A	N/A	ND	N/A	ND	N/A

ND: not detected; SD: standard deviation; RRT: relative retention time. RRT of Feno = 2.2.

**Table 3 pharmaceutics-14-00428-t003:** Bulk density, tap density, BET, compressibility, and flowability of Feno-loaded MS, Feno-loaded SDD, and mesoporous silica (Parteck^®^ SLC).

Sample ID	CBD (g/mL)	TD (g/mL)	BET (m^2^/g)	CPS (%)	ffc (Flow Remark [27])
20%-Feno-MS	0.53	0.65	N/A	15.12	23.18 (free flowing)
30%-Feno-MS	0.60	0.74	120.72	15.51	27.16 (free flowing)
30%-Feno-SDD-PVP	0.18	0.24	1.35	39.43	4.66 (cohesive)
Pure MS	0.42	0.48	432.24	19.57	12.27 (easy flowing)

## Data Availability

Not applicable.

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
