# Peer review of "Superiority of Mesoporous Silica-Based Amorphous Formulations over Spray-Dried Solid Dispersions"

_pharmaceutics, 2022, doi:10.3390/pharmaceutics14020428_

Round 1

Reviewer 1 Report

The authors addressed the reviewer's points in a satisfactory way, therefore the revised manuscript can be accepted for publication.

Reviewer 2 Report

The authors have included all requested corrections improving the manuscript content.

This manuscript is a resubmission of an earlier submission. The following is a list of the peer review reports and author responses from that submission.

Round 1

Reviewer 1 Report

This work presents a comparison between two different amorphous drug formulation methods, namely adsorption on mesoporous silica and spray drying in presence of matrix forming polymers.

The rationale of the study is sound, and the experimental methods are adequate for the task. There are, however, certain points that need consideration before the  manuscript can be accepted for publication:

  1. Although the conclusions drawn from this study are valid in this case, can it be assumed that they are universal, based only on a single example? Is the adsorption on mesoporous silica the best overall method for amorphous drug formulation? What are its limitations and drawbacks? In which classes of APIs does it work best?
  2. The authors should comment on the intermolecular interactions that are responsible for the superior stability of MS based formulations. It is somewhat surprising that there is no FTIR spectroscopic investigation of the mode of interaction.
  3. Page 9, line 217, X-ray diffractograms are not spectra.
  4. Figure 3. Are the graphs needed? The same information is contained in the Table.
  5. Figure 4. The X-axis scale shouldn't extent to 0 degrees, it doesn't contain any data up to 25 degrees.
  6. Figure 7. The burst effect in the case of SDD in SGF is explained by the rapid dissolution of surface adsorbed amorphous fenofibrate, which subsequently precipitates due to the high supersaturation. However, the authors should provide a convincing explanation for the persistent, increasing supersaturation in the case of the MS formulation in FaSSIF. What keeps fenofibrate from precipitating?

Reviewer 2 Report

The manuscript titled: Superiority of Mesoporous Silica Based Amorphous Formulation Over Spray-Dried Solid Dispersions by Liu and co-worker provides a comparison of fenofibrate based amorphous formulations using either mesoporous silica or polymer (PVP K90 or HPMC) as a carrier. The study is well designed and clearly written. It evaluates not only the phase of drug molecules in the formulation but also technological aspects of excipients that are frequently overlooked by other authors. While, I do like the manuscript in general there are copule of minor and one major point to address:

Minor:

  1. Materials and methods - please double check if the equipment used for the studies is specified in all subsections. For example, in 2.8.4 it is missing.
  2. SEM analysis - please provide a scale bar in Figure 2 images. Are authors confident SEM could distinguish between drug and silica crystals as both materials show very similar morphology? Do authors have an image of physical mixture of both materials?

Major:

The content of the drug in the materials as well as uniformity of content was not determined. With incipient wetness method it is likely that drug is not distributed uniformly within materials, also when using highly concentrated solutions of the drug in volatile solvents such as acetone crystallisation outside the pores happens quite often.

Could authors provide more details on the content of the loading solution and mesoporous silica in section 2.2?

Could authors determine content uniformity within the samples? For example, TGA analysis in the higher temperature range could give some idea of the organic content within the silicas.

What is the contact angle between the silica powder and loading solution? Is acetone good wetting agent for the powder? What is the powder permeability time for acetone?

PXRD analysis - for such complex solids the analysis time of ca. 10 minutes is not sufficient to detect low content of crystalline materials within silica. Also could authors make the resolution of PXRD plot better? Instead of multiple plots with separated y axis I would recommend having y axis common for all materials. Similarly in DSC plot y axis from -4.5 to 3 (W/g) (as well as use of the dashed line) makes data analysis difficult. Taking into account that maximum of 30 % is organic material and out of that as little as few percent can be crystalline it means that if taking 3 mg of the sample one will get signal from the transition of ca. 0.1 mg of the drug effectively. Furthermore, mesoporous silica has poor thermal conductivity that also makes DSC analysis of drug silica mixtures challenging.  While I am confident that incorporated drug stayed amorphous throughout the accelerated stability studies I am concerned of the encapsulation capability of the material.

While I do see some value in the presented work the manuscript do not provide sufficient novelty for its publication in Pharmaceutics in its current form.

Author Response

Please see the attachment, thanks.
